# Pierre Claverie: Weakening the Truth—A Catholic Post-Conciliar Model of Understanding Religious Plurality

**Alexandru-Marius Crișan** 

Research Unit for Intercultural Dialog (Research Center of Cultural Heritage and Socio-Cultural History), "Lucian Blaga" University of Sibiu, 550024 Sibiu, Romania; alexandru13marius@gmail.com

**Abstract:** Born into a *pieds-noir* family in Algeria in the first part of the 20th century, Pierre Claverie (1938–1996) realizes that he lived in a colonial bubble, completely ignoring the Algerian and Muslim realities. This "prise de conscience" will constitute the beginning of a deep mystical experience, a true process of "spiritual enlightenment" through which Claverie will try to re-establish himself in the lost meeting of his youth when he used to live in his "Western and Catholic bubble". His theological path will also be an institutional one: he will become a Dominican monk and Catholic bishop of Orano. Inevitably, his desire to spiritually encounter the Algerian reality in its Muslim identity will make Pierre Claverie reflect on the tension present in the paradox of the concept of *religious truth* (absolutely unique and/or multiple?). His reflection on the uniqueness/pluralism of religious truth represents a model of post-conciliar theological understanding that is worth examining and that supports a very current approach in today's religious and social world: understanding/accepting the truth of the other without the impression of betraying one's own truth. This study aims to deepen the mystical theological reflection of Bishop Pierre Claverie, considered a martyr in the Catholic Church, with regards to the concept of religious truth. It also to tries to understand how this reflection fits into the Catholic theological line inaugurated by the Second Vatican Council (1962–1965).

**Keywords:** Pierre Claverie; religious pluralism; religious truth; Second Vatican Council; inter-religious dialogue; weak thought; Gianni Vattimo

## 1. Introduction

In the early 1980s, the majority of Catholic Christians in Algeria had left the country and settled in France or other countries after Algeria's declaration of independence two decades earlier, causing a major change in the Catholic Church. The last few Christians who remained in Algeria were not just seen as a tiny minority but also as a symbol of the French invasion and colonization. Because of this, the Church in Algeria during those years needed to reframe the sense of its presence. Pierre Lucien Claverie, a Dominican monk and Bishop of Orano, was a man with a strong sense of faith who was able to illuminate this reflection toward a wide inter-religious opening.[1]

Pierre Claverie was born in 1938 in Algiers, when Algeria was still ruled by France. He was Bishop of Orano from 1981 until his martyrdom in 1996. Through his original thinking, Claverie tried to lead inter-religious reflection toward ideas such as otherness and religious plurality in an effort to reshape the concept of encounter. By doing so, he also reshaped the concept of religious truth. The years of Claverie's reflection coincide with those of the Second Vatican Council and of its immediate reception.

In this study, I will try to follow the dynamic of the concept of *religious truth* in Pierre Claverie's spiritual experience and theological reflection, with the precise purpose of understanding how Claverie's spiritual intuition could fit in—or not—with the theological line inaugurated by the Second Vatican Council. In other words, my final aim is to understand whether, and to what extent, the dialogical model of "Claverie" could be an option for those interested in inter-religious relations in today's Christian Churches (especially in

the Catholic Church, to which Claverie belonged), most of them communities where the concept of *exclusive truth* is still very much at the heart of the dogmatic faith teachings.

## 2. The Question of the Religious Pluralism and the Catholic Church in the 20th Century

All religions claim to be paths to salvation that (often) contain an exclusive truth. All propose a revelation of the mystery of God. Everyone, believer and non-believer alike, has at least at some time in their lives asked questions such as: Which is the true religion? Why so many religions? How can I relate to other religions if I am deeply rooted in my own faith and believe a certain religious truth? These are not easy questions to answer, but they confirm a universal human openness to transcendence and the need to be rooted in a stable truth (Di Tora 2014, pp. 260–61).

### 2.1. A General Overview of the Question of Religious Pluralism

There are different ways that scholars and theologians regard other religions and their function. When relating to this question from a neutral religious or/and abstract position(s), religious pluralism is somehow easier to understand as a matter of fact, and it mostly depends on qualities such as a religion's relevance for a society, its impact on personal development, its capacity to inspire certain qualities considered suitable today (humbleness, being capable of appreciating differences, open-mindedness, being tolerant, opposing proselytism, opposing anti-colonialist and anti-imperialist thinking). Instead, when trying to relate to religious pluralism from inside an assumed religious identity, two main aspects stand out: efficacy for salvation and truth. All religions pretend to be ways of salvation, and most of them claim to contain a revelation considered to be *the Truth*.

There are three main models to consider when speaking about religious diversity and the uniqueness or plurality of religious truth: exclusivism, inclusivism, and pluralism. In very short words, the lines of understanding swing from a total exclusivism that would mean that "all the other religions are wrong" (Di Tora 2014, p. 254)—underlying the existence of a unique truth possessed only by one religion—to a pluralism signifying that all religions are true, thus "entering in the salvation plan as totally independent" (Di Tora 2014, p. 254); in between, there is the inclusive view and a huge number of aspects to discuss.

Questions such as "Which religion is of truly divine origin and not only a human attempt to rich Divinity?" or "Which criteria should be taken into consideration when judging the truth of different religions?" (Fisichella 2007, pp. 39–40) very much reflect an exclusivist view of religious truth. First of all, we must underline the terminological problem of religious exclusivism. The term seemed to many as being too polemical and thus presenting negative connotations. Harold Netland and Veli-Matti Kärkkäinen mention the urge on the part of some specialists in the field to replace the term "exclusivism" with something more neutral such as *particularism* or *restrictivism* (Netland 2001, p. 46; Kärkkäinen 2003, pp. 80–81). But exclusivist views are quite normal in the early sources of most of religions. "Thus, for example, the earliest Buddhist and Christian sources prominently feature staunch criticisms of various rival teachings and practices as, respectively, false and useless or harmful" (Tuggy 1995). Followers of exclusivist views are present in each religion, since exclusivism is the easiest and the simplest form of approaching and understanding religious truth. The contemporary theologian Dale Tuggy observes that an "only one true religion" stance would be, at the end of the day, a *naïve exclusivism*:

> This position cannot be self-consistently maintained. Consider the claim that the cosmos was intentionally made. An informed Christian must concede that Jews and Muslims too believe this, and that they teach it as a central doctrine. Thus, if central Christian teachings are true, then so is at least one central teaching of these two rival religions. (Tuggy 1995)

The same inconsistency could be found when thinking about the implications for salvation in the exclusivist theories: how could one religion that preaches about a loving

God imagine this God of love sending the children who died and who were not-baptised or did not belong/practice a certain religion into hell, the same for mentally handicapped or righteous people living before the founding of a certain religion (Dupuis 2001, pp. 91–92)? An extreme exclusivist view would definitely be inconsistent, and it is rarely sustained as such; rather, an exclusivism with certain mentions and additional explanations or exceptions is more often present in theology and philosophy. The same applies for the opposite view: religious pluralism. Dale Tuggy speaks here also about *naïve pluralisms*. He mentions that one must truly be uninformed to believe that "all religions of some kind are the same in some valuable respect(s)" (Tuggy), this approach being truly rare, if not absent among scholars of any kind. Three directions must be mentioned when deepening the naivety of religious pluralism. Firstly, the so-called negative pluralism or *verificationist* pluralism. This means that all religions are equally false, simply because their truth cannot be empirically verified. The second type of naïve pluralism would be the idea that all religions would turn out to compose a final complementary truth (Tuggy). This is simply impossible because of the very different ways these religions understand Divinity, the human person, the world, and the final spiritual destiny of the cosmos. The third one would be the many shades of a more acceptable pluralism, such as the one proposed by the contemporary philosopher of religion John Hick (1922–2012). "The pluralist position advanced by John Hick has been and continues to be one of the most, if not the single most, significant and influential philosophical approaches to religious pluralism" (Norton 1995). The central claim of Hick is that there is only one ultimate divine Reality, who/which is, epistemologically speaking, unavailable to human beings in itself, but could nevertheless be experienced. Michael Barnes Norton explains very well the consequences of the pluralist position proposed by Hick:

> From here, he proceeds to his central claim that diverse religious traditions have emerged as various finite, historical responses to a single transcendent, ultimate, divine reality. The diversity of traditions (and the belief claims they contain) is a product of the diversity of religious experiences among individuals and groups throughout history, and the various interpretations given to these experiences. (...) in light of his epistemological arguments, Hick claims that all religious understandings of the Real are on equal footing insofar as they can only offer limited, phenomenal representations of transcendent truth. This position, which he calls the "pluralistic hypothesis," brings together elements of several philosophical perspectives on religion into a complex whole. (Norton 1995)

In between exclusivism and pluralism, there are different types of inclusivism, actually a more friendly or politically correct exclusivism or a less relativistic pluralism. Therefore, it is very difficult to differentiate the inclusivism of some more "soft" or better-argued forms of exclusivism or pluralism. These theological or philosophical directions maintain the superiority of a certain religion, also confessing that there are positive elements (which could be means of salvation or a partial truth) in other religions too. When speaking about eternal salvation, an inclusivist direction would always sustain the absolute centrality of an event/Person/God of a certain religion, even if salvation is accessible to non-members of that specific religion. One of the most well-known inclusivist theories is Abrahamic inclusivism, going back to the common background of the three related "Abrahamic" religions: Judaism, Christianity, and Islam. Even in the sphere of inter-Christian relations, an ecumenical inclusivism is most widespread and easily accepted; one confession maintains its claim to superiority or completeness of truth but seeks common ground or "elements of truth" in other confessions.[2]

*2.2. The Catholic Theology of the 20th Century on Religious Pluralism According to the Second Vatican Council*

The central event of the 20th century for the Catholic teaching is the Second Vatican Council, which tried to explore new ways of understanding religious pluralism and the question of religious truth.

*Nostra Aetate* addressed, for the first time, the question of religious pluralism in an official way and the attempt to have "a global approach" (Osto 2023, p. 180). The most relevant paragraph here would be the one in chapter two referring to the metaphor of one Sun with many rays:

> The Catholic Church rejects nothing that is true and holy in these religions. She regards with sincere reverence those ways of conduct and of life, those precepts and teachings which, though differing in many aspects from the ones she holds and sets forth, nonetheless often reflect a ray of that Truth which enlightens all men. Indeed, she proclaims, and ever must proclaim Christ "the way, the truth, and the life" (John 14:6), in whom men may find the fullness of religious life, in whom God has reconciled all things to Himself. (*Nostra Aetate* 1965, chapter 2)

In just a few lines, this paragraph contains different directions of understanding of the matter of religious pluralism. The Italian professor Giulio Osto underlines that "this well-known text is one of the most quoted paragraphs in the field of the theology of religions, but mostly without a suitable comment able to explain the main elements present in it" (Osto 2019, p. 111). The metaphor itself could be quite confusing, even if it can be traced to the Prologue of the Gospel of John. The image of *One Sun, many rays*, taken separately from the rest of text, would at a first glance send us towards a total pluralism of equal religions that contain the same revelation and truth (expressed in different ways). It is the mentioning of Christ and the quotation taken from John's Gospel that help us understand that it is Christ in the center of this metaphor, and through Him, "all men" are enlightened. The great theologian of the 20th century Pietro Rossano reads this paragraph in an inclusivist manner, putting the image of Christ-Pantokrator at the center of the world's salvation, despite all the religious diversity:

> It is in the light of Christ-Pantokrator that it is possible for us to theologically read the other religions, their diversity, their vitality and their persistence. (…) I don't want to say that all the religions contain different revelations of God, as if God would have revealed Himself in a way to a people, and in another way to another people. It is one thing the historical revelation received in Jesus Christ, and totally different the enlightening of the Logos Who enlightens every man. (Osto 2023, p. 177)

This principle is even clearer in the expression "The Catholic Church rejects nothing that is true and holy in these religions". *Lumen Gentium*, "the most authoritative of the Council's teaching on the dialogue with Muslims" (Olizar 2021, p. 146), also expresses a "what-is-common" understanding: "But the plan of salvation also includes those who acknowledge the Creator. In the first place among these are the Moslems, who, professing to hold the faith of Abraham, along with us adore the one and merciful God, who on the last day will judge mankind" (*Lumen Gentium* 1964, chapter 16).

Another interesting aspect derived from the analogical principle and observed by Isabel Olizar is that *Nostra Aetate*, despite mentioning the common aspects, "is silent, however, on essential aspects of the Muslim faith, namely on Muhammed and on the Qur'an" (Olizar 2021, p. 146). Olizar also quotes Georges Anawati, who writes: "it can be said the Council's Declaration gives an account, in the shortest possible form, on the Muslim theodicy, but not of the essence of the Moslem faith, which includes among its most important elements, belief in the prophetic mission of Mohamed" (Olizar 2021, pp. 146–47). This *silence* of the document is not something to condemn, but mostly something meant to build upon. Pietro Rossano draws attention to the possibly violent and dynamic aspects of a religious monotheist type of exclusivism: "social exclusion, war, jealousy, absolutism, all quite known to historians of religions" (Osto 2023, p. 168). Actually, in the third chapter, *Nostra Aetate* states:

> in the course of centuries not a few quarrels and hostilities have arisen between Christians and Moslems, this sacred synod urges all to forget the past and to work sincerely for mutual understanding and to preserve as well as to promote

together for the benefit of all mankind social justice and moral welfare, as well as peace and freedom. (*Nostra Aetate* 1965, chapter 3)

Therefore, if there is a *silence* regarding what divides—in this specific case—Christians from Muslims, it has a social purpose.

The *silence* could also be interpreted in a mystical way: as a safe space for an apophatic theology. Even if in *Nostra Aetate*, this aspect is not so evident; we can find it much more clearly in other documents of the Second Vatican Council. For example, chapter 7 in *Ad Gentes* (1965) speaks about "ways known only to God Himself" to lead to salvation "those inculpably ignorant of the Gospel". Another example would be *Gaudium et spes*:

The Christian man, conformed to the likeness of that Son Who is the firstborn of many brothers, received the first-fruits of the Spirit (Rom. 8:23) by which he becomes capable of discharging the new law of love. Through this Spirit, who is the pledge of our inheritance (Eph. 1:14), the whole man is renewed from within, even to the achievement of the redemption of the body (Rom. 8:23). (*Gaudium et spes* 1965, chapter 22)

Paragraphs like this, even if (or mostly because) Christ-centered, leave a space for the essential question of the apophatic mystical theology: is it what we know or what we do not know about God that is greater?

To conclude on the image of the religious pluralism found in *Nostra Aetate* and other documents of the Second Vatican Council, we could state that the documents are mostly coherent, developing a Catholic Christ-centered inclusivism named by Carmelo Dotolo in his *Teologia delle religioni* as an "asymmetric pluralism" or "the universality of the common minimum, where the openness towards the other must be conjugated with the absolute centrality of Christ" (Dotolo 2021, p. 163). He also writes about a mystical apophatic understanding of the silence of the documents on some thorny issues; Dotolo introduces the expression "eschatological reserve" (Dotolo 2021, p. 165): whatever cannot be solved or explained now will be solved in the eschaton.

## 3. Pierre Claverie—A Dialogical Model of Proclaiming a *Weak* Truth

*3.1. Encounter and Truth*

3.1.1. La Prise de Conscience of a Missed Encounter: A Spiritual Experience

The theological model for understanding the concept of religious truth developed by Pierre Claverie must be situated both in relation to his personal spiritual experiences and "to the evolution of the Catholic Church in Algeria during the second half of the twentieth century." (Olizar 2019, p. 163).

When speaking about the personal spiritual experiences of Claverie, it is absolutely clear that the basis for his entire dialogical vocation resides in the *prise de conscience* (the moment of awareness when, in his early 20s, he realized he had been living in a colonial and religious bubble until that moment); almost every paper dedicated to Pierre Claverie's life or theology sets the study's context apart from this particular event in his life. Whatever theological aspect is examined, whether inter-religious dialogue, religious reconciliation, or his mission or theological legacy as a whole, the studies dedicated to Claverie always depart from the words he himself wrote when, decades later, he appealed to the years of the *prise de conscience* in his autobiographical essay "Humanity in the Plural"[3]. This fact confirms the richness of spiritual and theological development evoked by that particular moment for Claverie.

The Algerian context of those years—Algeria still being under French rule—is very important. Phillip Naylor portrays the context as "a Manichaean society, divided by colonialism into an indifferent dualism marked, for example, by the physical separation of neighbourhoods (. . .). In particular, the other, the colonized, was excluded (. . .)" (Naylor 2010, p. 728). Being born in 1938[4], as the fourth generation of a French family, the young Pierre lived mainly in the European quarter of Algiers (Bab el Oued). Jean-Jacques Pérennès also describes the dramatic social reality of those years: "like most people in his social

sphere, (...) he lived until the age of twenty without meaningful contact with the Arabic and Muslim world next to his own" (Pérennès 2007b, p. 136). In the beginning, during Claverie's university years spent in Grenoble, in France, in the context of the Algerian quest for independence, the young Pierre was even tempted to join some extreme right movements (Pérennès 2019, p. 38), but soon, the *Other* prevailed in his spiritual landscape:

> I spent my childhood in the *colonial bubble*, in which relations between the two worlds were constantly absent. (...) The Other was ignored (...) Perhaps because I ignored the Other or because I denied his existence, one day he suddenly leapt right in front of me. He burst opened my shelter universe, which was ravaged by violence (...) and asserted his existence. (Claverie 2007, p. 258)

The spiritual need for encountering the *Other* thus became so important for Claverie's religious path that he would admit later: "the Other (...) became an overwhelming preoccupation. It is likely that my religious vocation stemmed from this" (Claverie 2007, p. 258). Even if "concurrently, Algeria and the Algerian Christianity were in the midst of the War of Liberation" (Naylor 2010, p. 729), the young Pierre Claverie, feeling that "Algeria is also his country" (Pérennès 2007b, p. 136), decided to go back in 1967, to what had already then become an independent Algeria. However, when Claverie came back and finally settled down in Algeria in 1967, he found a very different country (Pérennès 2007b, p. 136) from the one of his childhood and, most of all, a very different Catholic Church, a marginalized one, diminished and deserted because the French *pieds-noir* had left Algeria. The Church was seen as having played a colonial role during the French rule.

It is useful to try to understand also in a psychological context Claverie's moment of awareness as being linked, of course, to the colonizing and post-colonizing context of the Algeria of the times of his pastoral activity. Phillip Naylor, when describing the Algerian context, quotes a well-known psychiatrist—Frantz Fanon (1925–1961)—who speaks about the psychological shock and "the massive psychoexistensial complex" of the colonized nations, who were continuously "inferiorised psychologically and depersonalised existentially" by their colonisers (Naylor 2010, p. 727). My own understanding when looking at Claverie's *prise de conscience* is that what actually happened was a rare psychological process of an inverted colonizing shock. In other words, that "massive psychoexistensial complex" mentioned by Fanon passed from the colonized to the colonizer. But this fact was triggered by Pierre Claverie's Christian identity, which must have inspired in him a sense of guilt for the errors of his French fellows and the desire to pay for the "sins of the past" by sharing the condition of Algerians. From here came the desire to *algerise* himself and to encounter the *Other* in her/his full identity, including the aspect of the religious truth.

At a religious level, this spiritual-psychological event, the *prise de conscience*—as Claverie himself calls it in French—was the basis for the development of other concepts by our theologian, namely, the *colonial bubble* or the *colonial shell* and what I would call a *pilgrimage of decolonisation*. The *colonial bubble* will be renamed by Claverie as a *Catholic bubble*, *Christian bubble*, and even a *Western bubble* (Monge and Routhier 2020, p. 53). Claverie writes about the dynamic process of continuously coming out of different levels of isolation and seeking the encounter with the Other:

> I lived my youth in what I now call a *colonial shell* and discovered the extent to which people can effectively live in a shell without realising it. And because that shell has been broken under the pressure of the war in Algeria, I thank God and try to continually come out of other shells. Since then, I've rethought my beliefs, or at least I've searched my faith for something that could help me to open eyes, to open doors, to crack shells, to put people in touch, and finally, to do everything I can to prevent more tragedies like the ones I have I knew as a child. (Claverie 2015, pp. 17–18)

Claverie himself understands his *inner pilgrimage* of *de-isolation* as being made by following the steps of the biblical pilgrimages performed by Abraham, Moses, Virgin Mary, Apostle Paul, or even Jesus (Claverie 2015, p. 51 and further). According to the late Pope

Benedict XVI, Joseph Ratzinger[5], the idea of engaging in pilgrimage to meet the other as a way to one's the religious mindset, as presented in the Bible, has profound theological implications for the concept of religious truth. It signifies being able to "universalise the faith, by freeing it from particularity", moving away from "what is particular" and what could also be idolatrous (because it is a personal truth) towards "what is general, what is universal, what belongs to everyone" (ideas taken from: Ratzinger 1999, p. 253). With regards to the concept of religious truth, this signifies a process of *purifying one own's truth* by challenging it with what Carmelo Doto calls the "common universal truth of everyone" (Dotolo 2021, p. 162). Applying the ideas of Ratzinger and Dotolo to Claverie in this particular case, I could say that what Claverie attempted was actually an enlargement of the *Christian truth* (confessed by a minority in Algeria, *the particular truth* in the case of Algeria) through a free and profound encounter with the *Islamic truth* (confessed by the majority of Algerians, *the general truth* in the Algerian religious reality). In this sense, Pierre Claverie's intuition to purify the truth by exposing it to otherness comes very close to what Pascal has drawn attention to: the danger of making an idol out of the truth.[6] Pierre Claverie manages to overcome this danger.

### 3.1.2. The Other and His Own Truth

The first and most relevant category in the theological mind of Pierre Claverie that flows from his spiritual experience is the *Other*. Pierre Claverie decided that his spiritual mission would be not only to formally meet the other, but to develop a profound dialogue with the other's entire identity. It is relevant for our study to ask ourselves: why was the *Other* (the Algerian, the Muslim) ignored by those belonging to the colonial reality (Christian, Catholic, Western, European)? In his testimony, Claverie gives an answer: "I have never had Arabic friends (...) We were not racists, merely indifferent, ignoring the majority of the people in this country. They were a part of the landscape of our outings, the background of our meetings and our lives. They never were equal partners" (Claverie 1997, pp. 723–24). We could assume, from this answer, the main aspect that emerges is self-sufficiency. The "indifference" recalled by Claverie must be understood as a lack of interest for that which represented the identity of the victims, the colonized ones, including the religious aspect. There was no interest in the Algerian identity, including in the *religious truth* of the colonized Algerian people; of course, the opposite is also valid. Except for special cases, there was no interest on the part of the Algerians in the Christian religious identity of the French invaders. It is quite easy to understand this mutual approach of exclusion. First and foremost, the years alluded to by Claverie belong to the first half of the 20th century, decades before the Second Vatican Council; the pre-conciliar Catholic ethos used to be characterized by a strong reluctance towards any kind of religious pluralism. Secondly, the Islamic ethos, with its strong monotheistic aspect and absolute claim of possessing the full and final revelation, had a strict view on the uniqueness of the religious truth; if we add that the Catholic French identity was perceived in Algerian society as the religious identity of the invaders, having played a significant role in the colonial process, then we are able to quite clearly portray the image of mutual exclusion. The *other* one was truly *another one*: socially, culturally, and, most relevant for our study, religiously.

A significant part of Pierre Claverie's spiritual path was the attempt to break this wall of mutual indifference and exclusion, a wall reinforced by an anti-pluralistic religious ethos. This is why Claverie urged Christians and Muslims to see beyond the "thorny past" and beyond the categories of the past (Olizar 2021, p. 150):

> At this moment, the key word of my faith is *dialogue*, not because this is a strategic choice (...) but because I feel that dialogues constitute the relation of God with people and of people with each other. (...) May the other, may all others, be the passion and the wound through which God will be able to break into our fortress of self-satisfactory. (Claverie 1995, p. 36)

But Claverie's path will not stop at "a life dedicated to meeting the other" (Pérennès 2007b, p. 136). In his theological mind, the *Other* is intrinsically united with the religious

truth she or he professes. Or, in other words, in his understanding, Claverie unites the person with the professed truth. Izabel Olizar elucidates this important theme in Claverie's dialogical approach as "learning to recognize the *Other* as a subject" (Olizar 2021, p. 151). She writes that "Claverie wanted to move toward a religion that does not seek to exclude other forms of belief, but recognizes the other as a free and responsible subject" (Olizar 2021, p. 151). The *other* is united with her or his religious truth, and, thus, Claverie speaks about the *truth of the Others* (Claverie 2008, p. 330). He writes:

> I not only accept the Other is Other, a distinct subject with freedom of conscience, but I accept that she or he may possess a part of the truth that I do not have, and without which my own search for the truth cannot be fully realized. (see: Olizar 2021, p. 151)

Claudio Monge and Gilles Routhier, who tried to deepen the sense of hospitality in Claverie's theology, believe that he seems to be attracted to the idea of "differentiated truths" (Monge and Routhier 2020, p. 72). I do agree with this, but I find it incomplete and static, so I would go further and say that Claverie aims for something dynamic: not only admitting to, but accommodating a differentiated truth. If this is the case, it is no wonder that Claverie openly states, "I need the truth of the others" (Claverie 2008, p. 330), an expression that implies the dynamic aspect of a lively encounter. This approach gives originality to Claverie's understanding of religious pluralism with regard to the concept of truth, but it must be fully explained in order to avoid misunderstanding.

*3.2. Encounter verus Truth. A Weak Truth: Comparison with the "Pensiero Debole" Philosophical Thought of Gianni Vattimo*

Many times, Pierre Claverie mentions the need for an authentic encounter, having in mind all the above mentioned implications for the theological understanding of the truth. Some questions arise: Is Claverie not creating a tension in his theological thought between the concepts of *encounter* and *truth*? In other words: is Claverie not sacrificing or, at least, *weakening* the truth for the sake of the dialogue? Or, if the truth seems not to be worthy enough to keep us—Christians and Muslims—apart, is Claverie not appearing to be "a complete pluralist who did not think any single religion has greater truth or is more effective in apprehending the divine, not even his own?" (Crișan 2022, p. 8). In order to react to such doubts, we must understand the terms of a good encounter and the dynamic of religious truth with regard to the process of encountering otherness as according to Claverie.

In order to better understand Claverie's pluralist view on religious truth, I will start by deepening the theological relationship between the concepts of *encounter* and *truth* of Piere Claverie by comparing his thinking with the philosophical concept developed by the late Italian philosopher Gianni Vattimo[7] (developer of the so-called *pensiero debole*—weak thought).

The idea that stands at the basis of this form of thinking is that there is no longer a possibility of affirming the achievement of any stable or definitive truth (and it can be noted that there is a strong connection here with the problem of postmodernity). A particular and well-known writing in which Gianni Vattimo has had the occasion to develop his philosophical concept is the collective volume bearing exactly the name of the concept: *Il pensiero debole*, first published in 1983. Already in the preface, he makes a strong assumption: "the philosophical debate today has at least one point of convergence: there is no single, final, normative foundation." (Vattimo and Rovatti 1983, p. 7). Therefore, the concept of *weak thought* developed by Vattimo departs from a universal postmodern assumption: the historical period in which we live shows a certain doubt and weakening of the categories marked by the absolute (Woodward 2007, p. 180). That is to say, modernity has rejected any kind of absolute categories: an absolutist state, an absolute political philosophy, the idea of a general, absolute view on sexuality and gender, an absolute religious truth to be imposed in a certain social space, etc. Now, we live in a post-modern period that is marked, according to Vattimo's understanding, by what he calls *weakened thought* (See: Vattimo 1985,



pp. 138–53). In Vattimo's exact words, this means "a shift of the foundations' crisis within the idea of truth itself" (Vattimo and Rovatti 1983, p. 7). According to Dario Antiseri (born 1940), professor of Philosophy, who in his publications deepened Vattimo's thought, there are only two true pillars of the *pensiero debole* concept: the idea that humankind reads the world from within linguistic horizons and through categories that make evidence relative; and the idea according to which such categories are not fixed but historical (Antiseri 1993, p. 43). According to Vattimo, the categories through which we read reality lose access to "*ontos on*"—the very essence of being (Vattimo 1983, p. 22). This means that instead of having access to reality (the truth) itself, we only have access to personal and limited "experiences of the truth." (Antiseri 1993, p. 15). If *post-modernism* is an answer to the matter of historical time (*chronos*) for Vatimo, with regards to an empirical reason for the impossibility to impose an absolute truth, the late Italian philosopher speaks about the fact "that we have no access to things-in-themselves, only to appearances. (...) Thus, we can have no facts about things as they supposedly are in- themselves" (Woodward 2007, p. 181). As such, whatever we express as *truth* about any reality is limited to "language and interpretation" (Woodward 2007, p. 181).

Gianni Vattimo speaks about the many voices that are heard today, also because of modern means of communication. The concept of truth is understood in Vattimo's philosophical thinking not as a given external unity but as a social consensus built out of all the voices heard in modern day society. The plurality of voices is understood by Vattimo as being the remnants that result from the collapse of the concept of *absolute truths*. No claim to absolute truth in matters of religion, philosophy, political life, human anthropology, etc. can be imposed in our post-modern society after the collapse of *strong thought* (claiming to possess and proclaim an absolute truth), except in certain dictatorial states, by military force.

This is why Vattimo speaks about the tendency towards violence and about a *pluralismo-phobia* of those claiming to possess an absolute truth (political or religious). Gianni Vattimo mentions that the theory of the *weak truth* does at all not mean that there are no ultimate and general truths, such as physical or chemical rules that govern the cosmos or biological life.—he even admits the existence God, so of a religious truth—but these truths cannot be empirically demonstrated or generally imposed anymore in a post-modern society[8].

Let us now go back to Claverie and post-colonial Algerian society. Is there still a role for truth in the dialogical theology of Claverie that is dominated by the need for encounter? We must understand how the concept of truth was perceived in Algeria suddenly after the French conquest, and in his reflection, Claverie transforms the question of truth.

> The Christianisation of Algeria and its political colonization went hand in hand and many clerics believed that France had a providential mission to convert and civilise Africa. Relevant in this sense is the case of Archbishop Charles Martial Allemand-Lavigerie of Algiers (1825–1892). (...) For example, he organised orphanages for the many Algerian children orphaned by famines and epidemics of that period. As expected, most of those children converted to Catholicism. Another important step was the foundation of the Society of Missionaries of Africa, known as White Fathers (in 1868), and of the Missionary Sisters of Our Lady of Africa, known as White Sisters (in 1869), having the mission to "convert the Berbers". (Crișan 2022, p. 6)

From Archbishop Charles Martial Allemand-Lavigerie, who claimed to have had a divine and providential mission to preach—almost impose—Christianity in all Africa (the *colonial* or the *strong* truth)[9], to Bishop Pierre Claverie (the *weak* truth), there has been a process of *weakening* the concept of religious truth, very much like the process described by Vattimo. Claverie himself lived this *weakening* of the truth in his mentality. The *weakening* of the concept of truth in Claverie's understanding took place in several steps. First, he understood that imposing a religious truth by violence will lead, sooner or later, to the rejection of that truth, so the model of the colonial *truth* is destined to fail. He wrote: "on the day when a bishop set foot in this way in Algeria, the seeds of the war

of Independence and of the rejection of Christianity had already been sown" (Claverie 2015, p. 40). Or: "As soon as we claim—and in the Catholic Church we have had this sad experience during our history—to possess the truth or to speak in the name of humanity, we fall into totalitarianism and exclusion" (Claverie 2008, p. 330). Therefore, Claverie decided to abandon the categories of the past (colonial imposed truth). This aspect is seen also in Claverie's refusal to declare conversion as the final purpose of the Christian mission. His main biographer, Pérennès, speaks about "a renewed sense of the mission" (Pérennès 2007b, p. 140).

Secondly, Claverie manages to adopt a pluralist view on the truth but through a mystical understanding, thanks to his spiritual experience. Claverie's process of *weakening the truth* is thus totally particular, and it could not be characterized as a simple social pluralism of religious opinions. There is, though, something that links the two thinkers, Claverie and Vattimo: both emphasize that the *truth* we express (the religious one, in the case of Claverie) is only the result of a limited human experience of reality (Vattimo) or of the Divine (Claverie). The term *experience* is a keyword for both our thinkers. In Claverie's theological understanding, the *weakening* flows from a spiritual reflection and experience, from realizing that the infinity of God's glory is impossible to be fully perceived, and so fully *possessed*, by the human mind, and by consequence, there is no sense of maintaining human division because of different *truths*. "God is beyond any human representation, including the Christian one" (Claverie 2004, p. 81). From here to his pluralist statement "I need the truth of the others" is only a step, but a step made in the light of a spiritual Christ-centered reflection, as we can see from Pierre Claverie's words:

> I am a believer. I believe in one God, but I do not claim to possess that God, either through Jesus who reveals him to me, or through the tenets of my faith. One does not possess God. One does not possess the truth, and I need the truth of others. This is the experience that I am having now with thousands of Algerians in the sharing of an existence and the questions that we all ask ourselves. (Claverie 2008, p. 330)

Both Claverie and Vattimo accept that general, absolute truths are impossible to demonstrate empirically, and both of them disagree on the use of any violent constraint to impose a certain truth. Claverie is interested in the truth of faith and the impossibility of possessing the truth about Divinity; meanwhile, Vattimo emphasizes the idea of the impossibility of knowing the absolute truth to every kind of truth. On the one hand, in Vattimo, we find an absolute given empirically universal situation; on the other hand, in Claverie, we find the result of a religious reflection on the infinity of the Divine compared to human limitation.

Even if Pierre Claverie (born in 1938) and Gianni Vattimo (born in 1936) were contemporaries, Claverie never encountered the theory of the *pensiero debole*, and most probably, Vattimo never read or wrote about Claverie's reflections. I would nonetheless venture to assume that Claverie would have enjoyed the expression *weak truth*, because the spiritual discourse of the bishop of Orano is also built on the crucified image of the Messiah. I would almost use the expression "a weak Jesus Christ". Pierre Claverie decides to risk everything, including his life, in order just *to be* in Algeria, having absolutely no proselytist purpose, because he was inspired by the image of *a weak Christ*. "We are in Algeria because of this *crucified Messiah*" (Claverie 1997, p. 838), he would write, not long before his own martyrdom, confirming a mystical approach to the encounter with the other.

Repeating the question in the first lines of this subchapter: Did Claverie sacrifice or, at least, weaken his religious truth because of his dialogical theology that was dominated by the absolute need for encountering the other? The answer, as it comes out from the quotations cited, must be yes. The concept of truth comes second after the concept of *otherness* in Claverie's theological thinking. But, on the other hand, Claverie's pluralism does not totally fit with the philosophical-social model developed by Gianni Vattimo. Therefore, Claverie's dialogical understanding of the truth is not simply the weak thought developed by Vattimo and applied to the Algerian religious reality—even if, at first glance,

the result is quite similar. What distinguishes Claverie is a spiritual experience and the attention given to the *human person*. In Claverie, the truth of the other receives indirect attention: the person is in the first place, and her/his truth is relevant because it is almost an extension of the person.

Paradoxically, even if Claverie and Vattimo go against the idea of an absolute truth, their reflection had led both of them to affirming an absolute truth: the firm belief that "no single person possesses the truth. Everyone is looking for it" (Claverie 2008, p. 330).

## 4. Final Remarks: Pierre Claverie, a Post-Conciliar Model of Relating to Religious Plurality?

A comparison between the Second Vatican Council and Pierre Claverie's reflections is not really suitable for some significant reasons: their authority is not comparable. The conciliar documents are a common work conditioned by many factors, having to reach the entire world in its all diversity. Meanwhile, Claverie reflects freely; his theological thinking is available to us thanks to the publishing of sermons, spiritual retreats he gave, personal letters, etc. (Crișan 2022, pp. 15–16). Therefore, we speak about writings that were initially not thought to address a large number of readers. Last but not least, Claverie does not clearly intend to give an exegesis of the Second Vatican Council; he even criticizes some post-conciliar directions. What is truly feasible, instead, is to try to understand in which measure Pierre Claverie's reflection could be considered a post-conciliar theological model.

"Is Claverie's reflection a post-conciliar model?" I would venture to answer yes and no. Yes, given the matter of time (Claverie reflects and writes during and after the Second Vatican Council) and the matter of thematic compatibility (there is the same theological ethos of inter-religious openness). No, in the sense that Claverie's reflection flows directly from his own spiritual experience and follows a unique theological path of openness towards the truth of the other, considered to be worthy of attention not in itself but because the truth is almost an ontological extension of the other.

In the conclusion of the subchapter dedicated to the Second Vatican Council, we quoted the resumé made by Carmelo Dotolo of the views on religious pluralism present in the Council's documents: *an asymmetric pluralism*, *the universality of the common minimum*, and the presence of *silence regarding the thorny issues*. If we try to summarize, on the steps of Dotolo, Claverie's view on religious pluralism in a few short expressions, it would sound like this: *a large pluralism guaranteed by an apophatic approach to truth*, *the need to accommodate the entire truth of the other*, *overcoming and not hiding thorny issues for the sake of the encounter*. Out of this joining of ideas, one could suddenly observe that Pierre Claverie's theological reflection moves in a wider space from the very beginning, even if most of the categories of reflection are quite the same. Instead of having a "what is common attitude" or a Christ-centred inclusivism, Claverie prefers "facing our differences" (Olizar 2021, p. 152). "Claverie constantly rejected the emphasis placed on ideas such as a unique God, brothers in Abrahamic faiths, religions of the book, etc.—all of these being an exaggerated quest of underlining what is common for Jews, Christians and Muslims." (Crișan 2022, p. 7). The Secretariat for Non-Christians (later known as the Pontifical Council for Interreligious Dialogue), inaugurated in 1964, organized many interesting Islamic-Christian conferences, but Claverie never attended these large gatherings. In 1987, he was appointed part of the Pontifical Council for Interreligious Dialogue. This position "gave him the chance to temper the enthusiasm of many of those specializing in dialogue" (Pérennès 2007b, p. 138).

Our theologian's relaxed attitude towards the truth of the other, even if identical at an external level with what Gianni Vattimo described as the collapse of the claim of absolute truths for lack of full universal knowledge, nonetheless has two spiritual and theological catalysts. One is the idea that God's greatness is impossible to express in theological dogmatic expositions, so there is no sense in fighting over dogmatic truths. The second is that the other is sacred, and his own truth is united with him and must be respected. This very personal theological reflection and spiritual search cannot be truly compared and follows, as already mentioned, from other steps than those in the official theological

statements of documents of the Second Vatican Council. There is no empirical claim in Pierre Claverie, but rather a spiritual path. The Council of the Popes, Saints John XXIII and Paul VI, tries to deal with a given situation: the existence of many religions and how the official teaching of the Church could, at the level of the Magisterium, relate to this given situation.

In order to be really acceptable as an "orthodox" Catholic model, the theological reflection of Pierre Claverie will always need further explanation simply because, at first glance, he was not interested in drawing an obvious *Christological inclusivism* (even if Christ is central in his reflection, but in more discreet manner) based on the supposition of the given superiority of the Christian truth. The lack of Christological clarity could be explained by the fact that Claverie imagined his writings were addressed also to Muslim readers.

Before concluding, we must admit that a yes or no conclusion is impossible to express without ignoring the theological complexity of the Algerian historical and religious context, Claverie's reflection, and post-conciliar Catholic theology. Even if not directly theologically influenced, Pierre Claverie's life during and after the Second Vatican Council offered him the freedom to reflect. The Council inaugurated not only a renewed theological direction to follow, but mostly an ethos of openness that inspired and confirmed many in their theological reflections. This post-conciliar ethos is present in Claverie's writings, and he took full advantage of it.

**Funding:** This work was supported by a Hasso Plattner Excellence Research Grant (LBUS-HPI-ERG-2020-XX), financed the Knowledge Transfer Center of the Lucian Blaga University of Sibiu.

**Conflicts of Interest:** The author declares no conflict of interest.

## Notes

[1] For more information on the religious Algerian context before and during Claverie's activity, see: in English (Crișan 2022, pp. 5–7) or in French (Crișan 2023, pp. 21–23).

[2] See, for example, the case of the Second Vatican Council (Mortola 2019) or the recent inter-Orthodox discussion during the Council of Crete (Crișan 2020).

[3] See, for example, the subchapters: "A Life Dedicated to Meeting the Other" (Pérennès 2007b); the subchapter "Claverie's Prise de conscience" in Phillip C. Naylor's study (Naylor 2010); the subchapter "Pierre Claverie of Algeria. La Prise de conscience—A Call for Other Kind of Mission" in (Crișan 2022).

[4] We do not intend to present the biography (or bibliography) of Pierre Claverie in this study; that is already available in many studies. For a more detailed biography, see the most important writing of his main biographer, the French original (Pérennès 2000) or the English translation (Pérennès 2007a). For updated bibliographical information, see (Crișan 2022, pp. 2–3).

[5] Actually, an author that does not always share Claverie's intuition on religious pluralism.

[6] The exact words of Blaise Pascal: "On se fait une idole de la vérité même; car la vérité hors de la charité n'est pas Dieu, et est son image et une idole", "We make an idol of truth itself; for truth apart from charity is not God, but His image and an idol". See: (Pascal 1897, p. 130).

[7] Gianteresio Vattimo (1936–2023), Italian philosopher who graduated in Turin and Heidelberg. His philosophy can be characterized as post-modern, thanks to his original emphasis on the *pensiero debole* (*weak troght*). He received several awards from different universities for his notable and original ideas.

[8] I would say that this theory was verified quite well during the last years of the Coronavirus pandemic: a uniform view on the nature of the virus or the efficiency of the vaccination failed to be generally accepted, even if we speak about a scientific theory demonstrated by competent medical and scientifical authorities.

[9] See, for example, some of his statements: "Algeria is only the door opened by Providence on a barbaric continent of 200 million souls. It is especially there that we must bring the Catholic apostolate." Or: "We cannot leave these people with their Qur'an. France must give them the Gospel". Cf. (Naylor 2010, p. 725).

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
