# Peer review of "Pierre Claverie: Weakening the Truth—A Catholic Post-Conciliar Model of Understanding Religious Plurality"

_religions, doi:10.3390/rel14121462_

Round 1
Reviewer 1 Report
Comments and Suggestions for Authors
The contribution presents three aspects of great originality. First, Claverie's placement in dialogue with the teachings of the Second Vatican Council. Secondly, the essential presentation of the main nodes of Claverie's attitude. Thirdly, the original comparison with the thought of Vattimo
The author can add more bibliography on Gianni Vattimo, even including some precise references to a particular work, so as to give more motivation for his arguments.
Author Response
Thank you for your review and suggestion!
I researched and tried to identify one of the most quoted writings of Vattimo when speaking about the pensiero debole. I named it as you recommended "a particular work". More quotations from this book were inserted. For better explanation I also inserted some quotations from a professor known to be an "exeget" of Vattimo's pensiero debole. I have also added some phrases meant to better explain my view on what brings together and what differentiate the two thinkers.
I have now uploaded both a clean version and a track-changes one.
Thank you!

Reviewer 2 Report
Comments and Suggestions for Authors
The article is well-balanced. The introduction shows the context in which arise the point of view of the author considered. The development of the text manages to show the importance of Claverie's thought about a conception of truth which allows to the construction of an interreligious dialogue.
Author Response
Thank you for your review!
Some minor improving attempts:
I inserted quotations from one of the most mentioned writings of Vattimo when speaking about the pensiero debole. For a clearer understanding, I also inserted some quotations from a professor known to be an "exeget" of Vattimo's pensiero debole.
I have also added some phrases meant to better explain my view on what brings together and what differentiate the two thinkers when speaking truth and inter-religious dialogue.
I have now uploaded both a clean version and a track-changes one.
Thank you!

Reviewer 3 Report
Comments and Suggestions for Authors
I find the article very interesting. I knew the figure and read Perennes' biography with great interest. I know personally people who were collaborators of the bishop. I am also rather well informed on the accademic side of the discussion on exclusivism, inclusivism and pluralism. The author uses other references than i am used to, in part. The main merit is the introduction of Vattimo's pensiero debole. Well done. The overall conclusion sounds rigt for me. And it proves for me that this magnificent figure was more a man of the field, a pastoral figure, however with his serious scholarly dominican background he was surely one of the best on the field. But I see now better his was not a professionnal theologian, by temperament, I think so the theological possibilities his position could disclose came not completely to maturity. If he had lived longer maybe ..., but a theological temperament is a rare gift Claverie didn't possess, in this sense Geffré of Coda brought us farer on the way where Claverie did some interesting steps in the right direction, as the article underlines.
Author Response

(The authors gave the same response as above.)
